# Resource-Constrained Federated Continual Learning: What Does Matter?

**Yichen Li**[1,2]**, Yuying Wang**[3]**, Jiahua Dong**[2]*****Haozhao Wang**[1]**,**
**Yining Qi**[1]**, Rui Zhang**[1]†****Ruixuan Li**[1]*****
[1]School of Computer Science and Technology,
Huazhong University of Science and Technology, Wuhan, China
[2]Mohamed bin Zayed University of Artificial Intelligence, Abu Dhabi, United Arab Emirates
[3]School of Computer Science and Technology, Soochow University, Suzhou, China
{ycli0204,hz_wang}@hust.edu.cn, rayteam@yeah.net

## Abstract

Federated Continual Learning (FCL) aims to enable sequential privacy-preserving model training on streams of incoming data that vary in edge devices by preserving previous knowledge while adapting to new data. Current FCL literature focuses on restricted data privacy and access to previously seen data while imposing no constraints on the training overhead. This is unreasonable for FCL applications in real-world scenarios, where edge devices are primarily constrained by resources such as storage, computational budget, and label rate. We revisit this problem with a large-scale benchmark and analyze the performance of state-of-the-art FCL approaches under different resource-constrained settings. Various typical FCL techniques and six datasets in two incremental learning scenarios (Class-IL and Domain-IL) are involved in our experiments. **Through extensive experiments amounting to a total of over 1,000+ GPU hours, we find that, under limited resource-constrained settings, existing FCL approaches, with no exception, fail to achieve the expected performance.** Our conclusions are consistent in the sensitivity analysis. This suggests that most existing FCL methods are particularly too resource-dependent for real-world deployment. Moreover, we study the performance of typical FCL techniques with resource constraints and shed light on future research directions in FCL.

## 1 Introduction

Federated Learning (FL) emerges as a distributed learning paradigm, facilitating the collaborative training of a global deep learning model among edge clients while ensuring the privacy of locally stored data [38, 54, 53, 26]. Recently, FL has garnered significant interest and found applications in diverse domains, including recommendation systems [61, 7, 25, 24] and smart healthcare solutions [59, 41, 71].

Typically, FL has been actively studied in a static setting, where the number of training samples does not change over time. However, in a realistic FL application, each client may continue collecting new data and train the local model with streaming tasks, leading to performance degradation on previous tasks [60, 27, 28]. Such a phenomenon is known as catastrophic forgetting [9] in the Continual Learning (CL) paradigm. This challenge is further compounded in FL settings, where the local data remains inaccessible to others, and training on clients is constrained by limited resources.

---

*Jiahua Dong and Ruixuan Li are corresponding authors.
†Homepage: https://www.ruizhang.info/

39th Conference on Neural Information Processing Systems (NeurIPS 2025).

Table 1: Primary Directions of Progress in FCL. Analysis of three kinds of classifications in the study of the CL system (replay, regularization, and network), with **bold** highlighting the main contribution. Besides, we further summarize four typical FCL techniques (**S**ample **C**aching, **D**ata **S**ynthesis, **K**nowledge **D**istillation, and Network Extension) from existing works and focus on two common FCL scenarios. Here "Constraint" denotes the research based on whether the additional resource overheads are required (**M**emory **B**uffer, **C**omputational **B**udget and **L**abel **R**ate).

| Dir. | Reference | Contribution | Scenarios | | Constraints | | | Typical FCL Techniques | | | |
|---|---|---|---|---|---|---|---|---|---|---|---|
| | | | Class-IL | Domain-IL | MB | CB | LR | SC | DS | KD | NE |
| | Centralized | | ✓ | ✓ | ✗ | ✗ | ✓ | ✗ | ✗ | ✗ | ✗ |
| | FedAvg [38] | | ✓ | ✓ | ✗ | ✗ | ✓ | ✓ | ✗ | ✗ | ✗ |
| | FedProx [22] | | ✓ | ✓ | ✗ | ✗ | ✓ | ✓ | ✗ | ✗ | ✗ |
| Replay | TARGET [66] | **Exemplar Sample** | ✓ | ✗ | ✓ | ✗ | ✓ | ✗ | ✓ | ✓ | ✗ |
| | FedCIL [45] | **Generation & Alignment** | ✓ | ✗ | ✓ | ✓ | ✓ | ✗ | ✓ | ✓ | ✗ |
| | Re-Fed [23] | **Synergistic Replay** | ✓ | ✓ | ✓ | ✗ | ✓ | ✓ | ✗ | ✗ | ✗ |
| | AF-FCL [58] | **Accurate Forgetting** | ✓ | ✗ | ✗ | ✓ | ✓ | ✗ | ✓ | ✓ | ✗ |
| Regularization | GLFC [5] | **Class-Aware Loss** | ✓ | ✗ | ✓ | ✗ | ✓ | ✓ | ✗ | ✓ | ✗ |
| | FOT [2] | **Orthogonality Projection** | ✓ | ✗ | ✗ | ✓ | ✓ | ✗ | ✗ | ✗ | ✓ |
| | CFeD [37] | **Knowledge Distillation** | ✓ | ✓ | ✓ | ✓ | ✓ | ✗ | ✗ | ✓ | ✗ |
| | FedET [35] | **Pre-training Backbone** | ✓ | ✗ | ✓ | ✗ | ✓ | ✓ | ✗ | ✓ | ✓ |
| | MFCL [1] | **Data-Free Distillation** | ✓ | ✗ | ✓ | ✓ | ✓ | ✗ | ✓ | ✓ | ✗ |
| | LGA [4] | **Category-Aware Loss** | ✓ | ✗ | ✓ | ✗ | ✓ | ✗ | ✓ | ✗ | ✓ |
| Network | FCL-BL [19] | **Broad Learning** | ✓ | ✗ | ✗ | ✓ | ✓ | ✓ | ✗ | ✗ | ✓ |
| | FedWeIT [62] | **Parameter Decomposition** | ✓ | ✗ | ✗ | ✗ | ✓ | ✗ | ✗ | ✗ | ✓ |
| | pFedDIL [30] | **Knowledge Matching** | ✗ | ✓ | ✗ | ✗ | ✓ | ✗ | ✗ | ✓ | ✓ |

To address this issue, researchers have studied federated continual learning (FCL), which enables each client to learn from a local private and incremental task stream continuously. One mainstream technique is to re-train the model with cached or synthetic samples later, a strategy referred to as data replay. FedCIL is proposed in [45] to reconstruct previous samples with a learnable generative network for replay, improving the retention of previous information. The authors in [23] propose to discern important samples from streaming tasks and update the local model with both cached samples and the current task's samples. Another promising solution for FCL is to limit the update of model parameters to new tasks, a process known as parameter regularization. The works in [5, 4] concentrate on federated class-incremental learning, addressing scenario-specific challenges through the computation of supplementary class-imbalance losses to train a unified global model. It is studied in [2] that projecting the different tasks' parameters onto orthogonal subspaces prevents new tasks from overwriting previous task parameters. In addition, the authors in [62, 19, 65] extend the backbone model with multi-head classifiers to isolate the task-specific parameters to prevent forgetting.

However, the current FCL literature overlooks a key necessity for practical real deployment of such algorithms. In particular, most existing methods focus on offline federated continual learning where, despite limited access to previous data, training algorithms do not have restrictions on the training resources. For example, Snapchat [68], in 2017, reported an influx of over 3.5 billion short videos daily from users across the globe. These videos had to be instantly processed for various tasks, from image rating and recommendation to hate speech and misinformation detection. Otherwise, new streaming data will accumulate until training is completed, causing server delays and worsening user experience. Different from the conditions in the centralized server, distributed devices (usually mobile or edge devices) pose greater challenges to the training resources. In this paper, we mainly focus on the following three mainstream resources, which are expensive on distributed devices.

**Memory Buffer.** Memory buffer serves as a crucial training resource for many existing methods, particularly those based on replay, as they necessitate the storage of previous samples or synthetic data for retraining. Additionally, some other methods relying on regularization also require buffering a portion of sample data or utilizing extra auxiliary datasets to facilitate the training process. In recent years, despite the advanced maturity of current storage technology, numerous platforms possess vast storage capacities, exemplified by computer SSDs priced at roughly 0.061\$/GB in the market[3].

---

[3]Price reference for PC: *https://www.amazon.com/ssd/s?k=ssd*

Nevertheless, when it comes to distributed devices, augmenting storage capacity remains costly, as seen in the iPhone's storage price, hovering around 0.487\$/GB — nearly eightfold the price of computer SSDs[4]. Therefore, a thorough examination of the memory buffer overhead demanded by existing FCL methods is imperative.

**Computational Budget.** The computational budgets of continual learning (CL) algorithms tend to exceed those of static data-based methods, as the model must converge on each incremental task prior to learning new ones. Amidst the increasing influx of streaming data, it is paramount for each client to optimize their computational efficiency. However, distributed devices (e.g., IoT and mobile devices) are usually equipped with portable hardware and sacrifice powerful computing resources. A feasible way for distributed devices to improve computational efficiency is to rent cloud platform computing resources, but this is expensive. According to [44], executing a CL algorithm on the CLEAR benchmark [34], which performs 300,000 iterations, incurs approximately a cost of 100\$ on an A100 Google instance, equating to 3\$ per hour for a single GPU. However, in existing methods, such as those based on replay, additional computational budgets are required for retraining the cached samples or training generative models to produce pseudo-data. On the other hand, regularization-based methods may necessitate extra computation for knowledge distillation. In FCL scenarios, limiting the computational budget is necessary for reducing the overall cost.

**Label Rate.** Additionally, the majority of existing works assume that a full set of labels is accessible. Only recently, some work in the centralized server started to consider budgeted continual learning, which aims to ensure the applicability of continual learning algorithms under real-world scenarios. Nevertheless, existing FCL methods still concentrate on a fully labeled data stream, while the data may be presented in an unlabeled form, and the human annotation budget is always large. Although existing FCL methods cannot be trained directly on unlabeled datasets, it is meaningful to appropriately reduce the label rate to observe the impact of the number of training samples on the performance of these methods.

This raises the question: "*Do existing federated continual learning algorithms perform well under resource-constrained settings?*" To address this question, we exhaustively study federated continual learning systems, analyzing the effect of the primary directions of progress proposed in the literature in the setting where algorithms are permitted to limit all three training resources. We evaluate and benchmark at scale various existing FCL methods and summarize four typical FCL techniques (Sample Caching, Data Synthesis, Knowledge Distillation, and Network Extension) that are common in the literature. Evaluation is carried out on six large-scale datasets in two scenarios (Class-IL and Domain-IL), amounting to a total of 1,000+ GPU hours under various settings. As shown in Table 1, we conclude the existing FCL methods and analyze their main contributions and typical techniques with training resources involved. We summarize our empirical conclusions in three folds:

- First, we are the first to revisit resource constraints in federated continual learning and analyze how all three training resources (memory buffer, computational budget, and label rate data) can pose a great challenge to the federated continual learning issue.

- Then, we conduct extensive experiments on more than ten existing FCL algorithms with different limited resources and find that existing FCL literature is particularly suited for settings where memory is limited and less practical in scenarios having limited computational budgets and sparsely labeled data.

- Finally, we analyze the effect of typical FCL techniques with resource constraints and discuss the future research directions of federated continual learning with resource constraints.

## 2 Dissecting Federated Continual Learning Systems

Federated continual learning methods typically propose a system of multiple components that jointly help improve learning performance. The problem formulation is illustrated in Appendix B. In this section, we analyze FCL systems and dissect them into their underlying techniques. This helps to analyze and isolate the role of different components with different techniques under our resource-constrained settings and helps us to understand the most significant techniques. Overall, we summarize four major techniques from main contributions in Table 1.

---

[4]Price reference for Phone: *https://www.apple.com/shop/buy-iphone/*

(1) *Sample Caching.* Rehearsing samples from previous tasks is a basic approach in FCL. Particularly when access to previous samples is restricted to a small memory, they are used to select which samples from the stream will update the memory. Recent FCL works in [23, 29] propose to synergistically cache samples based on both the local and global understanding. Other methods have not considered the sample selection strategy in FCL. For a fair comparison in our constrained memory buffer setup, we employ random sampling equally to corresponding methods.

(2) *Data Synthesis.* Considering the security and privacy of data in some scenarios, or data protection under GDPR, the data of previous tasks cannot be cached locally. Some studies use generative models to learn the distribution of previous tasks and generate synthetic data for data replay. The authors in [45, 66] employ the GAN model [42] to generate synthetic samples while [58] explore the NF model [48] to learn the data distribution accurately. In our setting, the synthetic data will also occupy the memory buffer and the additional training cost brought by the generative model will share the computational budget with the target model.

(3) *Knowledge Distillation.* As a popular approach, knowledge distillation preserves model performance on previous tasks by distilling the knowledge from the old model (teacher) to the new model (student). In the FCL, the distillation can be done on either the client side or the server side. On the client side, the student model can be the current model, while the teacher model is the one that has been trained for many previous tasks. On the server side, the student model is always the global model and the teacher model can be denoted as the ensemble prediction by each participating client. In this paper, we only focus on the distillation done on the client side since the resources of the server are not so scarce. The distillation will bring an extra computational budget compared with the basic FedAvg. Moreover, additional distillation data used on the client side will also be included in the memory buffer overhead.

(4) *Network Extension.* Network extension strategies have been used for two objectives. On the one hand, it has been hypothesized that the large difference in the magnitudes of the weights associated with different classes in the last fully connected layer is among the key reasons behind catastrophic forgetting. There has been a family of different methods addressing this problem [62, 35, 2]. On the other hand, several works attempt to adapt the model architecture according to the data. This is done by only training part of the model or by directly expanding the model when data is presented [16, 65]. However, most of the previous techniques in this area do not apply to our setup. Most of this line of work [62, 35, 2], assumes a task-incremental setting, where test samples are known to what set of tasks they belong with known boundaries between streaming tasks. For a fair comparison, we modify these methods to make a correct classification among all classes from streaming tasks.

## 3 Experiments

We first start by detailing the experimental setup, datasets, three training resources, and evaluation metrics for our large-scale benchmark. We then present the main results of an evaluation of various FCL methods, followed by an extensive analysis. Our experiments are designed to answer the following research questions that are of importance to practical deployment of FCL methods, while also pointing out the future research directions in FCL.

- *How do resource constraints affect the performance of existing FCL methods?* (Section 3.2.1)

- *How do typical techniques studied in FCL literature work under different resource-constrained settings?* (Section 3.2.2)

### 3.1 Experiment Setup

#### 3.1.1 Datasets

We conduct our experiments with heterogeneous datasets over two typical scenarios: Class-Incremental Learning and Domain-Incremental Learning on six datasets: CIFAR-10 [18], CIFAR-100 [18], Tiny-ImageNet [20], Digit-10, Office-31 [50] and Office-Caltech-10 [69]. More details can be found in Appendix B.1.1.

Table 2: Performance comparison of various methods in two incremental scenarios w.r.t. sufficient resources.

| Method | CIFAR-10 | | CIFAR-100 | | Tiny-ImageNet | | Digit-10 | | Office-31 | | Office-Caltech-10 | |
|---|---|---|---|---|---|---|---|---|---|---|---|---|
| | $A(f)$ | $\bar{A}$ | $A(f)$ | $\bar{A}$ | $A(f)$ | $\bar{A}$ | $A(f)$ | $\bar{A}$ | $A(f)$ | $\bar{A}$ | $A(f)$ | $\bar{A}$ |
| FedAvg [38] | 40.94±1.32 | 54.33±1.03 | 25.14±0.87 | 35.97±1.12 | 32.87±0.45 | 48.55±0.34 | 80.13±0.37 | 90.59±0.31 | 45.82±0.58 | 47.65±0.47 | 50.55±0.63 | 55.36±0.51 |
| FedProx [22] | 40.10±0.92 | 53.52±0.73 | 25.23±0.71 | 35.83±0.83 | 29.92±0.67 | 45.12±0.53 | 81.35±0.44 | 91.73±0.39 | 46.14±0.61 | 49.90±0.54 | 51.66±0.59 | 56.29±0.44 |
| FedAvg+ER [49] | 42.33±1.54 | 55.15±1.29 | 27.07±0.84 | 37.77±1.06 | 33.31±0.52 | 48.56±0.49 | 80.48±0.30 | 90.86±0.27 | 46.72±0.53 | 49.50±0.42 | 52.62±0.57 | 56.16±0.51 |
| FedProx+ER [49] | 41.07±1.07 | 54.53±0.98 | 26.97±0.78 | 36.10±0.96 | 34.08±0.79 | 45.54±0.66 | 82.06±0.49 | 92.68±0.51 | 46.21±0.73 | 51.66±0.67 | 53.58±0.68 | 57.18±0.56 |
| FedCIL [45] | 45.35±1.76 | 56.54±1.35 | 24.88±1.44 | 34.70±1.08 | 28.96±0.90 | 44.54±0.71 | 85.09±0.91 | 92.83±0.68 | 48.78±0.84 | 50.24±0.72 | 54.69±1.03 | 58.15±1.17 |
| TARGET [66] | 43.78±1.67 | 56.34±1.38 | 24.01±1.33 | 33.38±0.87 | 29.14±0.70 | 45.02±0.54 | 84.73±1.06 | 93.11±0.88 | 48.29±1.17 | 50.32±0.85 | 53.19±0.82 | 55.99±0.73 |
| AF-FCL [58] | 44.95±1.39 | 57.09±1.19 | 24.62±0.93 | 34.60±0.88 | 27.15±1.12 | 43.58±0.74 | 85.99±0.81 | 93.49±0.54 | 49.07±0.74 | 50.87±0.61 | 54.12±1.06 | 56.94±0.77 |
| Re-Fed [23] | 43.44±0.51 | 57.52±0.45 | 27.56±0.31 | 37.82±0.11 | 35.99±0.41 | 52.19±0.27 | 84.11±0.36 | 94.33±0.29 | 49.53±0.36 | 52.32±0.30 | 55.08±0.49 | 58.82±0.24 |
| GLFC [5] | 43.08±0.76 | 55.02±0.68 | 26.69±0.23 | 36.11±0.18 | 34.73±0.56 | 47.37±0.31 | 79.15±0.43 | 90.46±0.19 | 45.83±0.23 | 47.31±0.20 | 52.37±0.34 | 54.86±0.15 |
| FOT [2] | 43.74±1.02 | 58.24±0.78 | 28.43±1.14 | 38.57±0.95 | 34.23±0.74 | 49.52±0.61 | 82.97±0.67 | 90.40±0.55 | 49.35±0.98 | 52.09±0.81 | 52.18±0.53 | 55.11±0.60 |
| CFeD [37] | 46.54±1.13 | 58.23±1.78 | 28.37±0.46 | 35.43±0.86 | 33.05±0.30 | 47.98±0.62 | 80.23±0.11 | 89.13±0.22 | 44.26±1.69 | 46.78±1.64 | 50.13±0.53 | 54.89±0.92 |
| FedWeIT [62] | 41.52±1.11 | 54.65±0.95 | 26.02±0.94 | 36.38±0.72 | 34.13±0.51 | 49.24±0.46 | 81.65±0.58 | 91.37±0.30 | 46.72±0.63 | 48.83±0.44 | 51.97±0.27 | 56.40±0.19 |

### 3.1.2 Baselines

In this paper, we follow the same protocols proposed by [38, 47] to set up FIL tasks. We evaluate our resource-constrained settings with twelve baselines: FedAvg [38], FedProx [22], FL+ER[49], FCIL [5], FedCIL [45], Target [66], AF-FCL [58], Re-Fed [23], FOT [2] and FedWeIT [62]. More details about each baseline can be found in Appendix B.1.2.

### 3.1.3 Resource Constraints

We analyze three common constrained resources at edge devices: Memory Buffer, Computational Budget, and Label Rate. Here, we abbreviate the memory buffer size as $M$, which means the number of the cached samples in the edge storage, and the computational budget amount as $B$ that denotes the gradient step during the model update; Label Rate as $R$, which represents the ratio of labeled data and unlabeled data. We have adopted different restriction methods for memory buffers and computational budgets for different baselines. The memory buffer is used to cache samples from previous tasks (Re-Fed[23], FCIL[5]), synthetic data (FedCIL[45], Target[66], AF-FCL[58]), and distillation samples (CFeD[37]). The computational budget is shared by the generative model (FedCIL[45], AF-FCL[58]), personalized model (Re-Fed[23]), knowledge distillation (FOT[2], CFeD[37]). The label rate is equally employed to all baselines, which controls the ratio of the labeled data.

### 3.1.4 Configurations

Unless otherwise mentioned, we use ResNet18 [10] as the backbone model for most methods and use the Dirichlet distribution $\text{Dir}(\alpha)$ to distribute local samples to stimulate data heterogeneity for all tasks where a smaller $\alpha$ indicates higher data heterogeneity. Here we report the final accuracy $A(f)$ when the model finishes the training of the last streaming task and the average accuracy $\bar{A}$ across all streaming tasks. Although we try to eliminate the gap between baseline methods other than the core algorithm design in the experiment, there are still some methods that directly depend on the backbone model itself. In our experiments, FedCIL[45] employs ACGAN for generative replay, and AF-FCL[58] needs to use a Normalized Flow model to generate the feature space for an accurate forgetting. In addition, we try to adopt the ResNet18 to the FOT[2] but achieve an unstable result. According to the available code in the original manuscript, we use AlexNet as the backbone model for the FOT[2]. The feasibility of using the ResNet network and achieving excellent performance through simple parameter tuning in the FOT algorithm is still under investigation.

**Hardware.** Each experiment setting is run twice and we take each run's final 10 rounds' accuracy and calculate the average value and standard variance. We use Adam as an optimizer with a linear learning rate schedule. We set the remaining parameters according to the values in the original open-source code, such as the weight of multiple distillation losses. All experiments are run on 8 RTX 4090 GPUs and 8*2 RTX 3090 GPUs.

## 3.2 Overview Performance under Resource Constraints

In this section, we systematically investigate the influence of resource constraints on existing FCL methods and typical techniques studied in the FCL literature.

Table 3: Performance comparison of various methods in two incremental scenarios w.r.t. extremely limited resources.

| Method | CIFAR-10 | | CIFAR-100 | | Tiny-ImageNet | | Digit-10 | | Office-31 | | Office-Caltech-10 | |
|---|---|---|---|---|---|---|---|---|---|---|---|---|
| | $A(f)$ | $\bar{A}$ | $A(f)$ | $\bar{A}$ | $A(f)$ | $\bar{A}$ | $A(f)$ | $\bar{A}$ | $A(f)$ | $\bar{A}$ | $A(f)$ | $\bar{A}$ |
| FedAvg [38] | 10.11±1.11 | 21.76±0.78 | 4.88±0.79 | 12.37±1.12 | 16.11±0.65 | 30.99±0.41 | 72.00±0.45 | 80.57±0.27 | 8.42±0.73 | 8.97±0.51 | 9.65±0.35 | 10.13±0.12 |
| FedProx [22] | 10.04±1.30 | 21.62±1.23 | 4.59±1.70 | 11.44±1.52 | 16.92±0.83 | 31.24±0.57 | 71.68±1.14 | 80.85±0.85 | 8.98±0.74 | 9.45±0.39 | 9.37±0.77 | 9.84±0.42 |
| FedCIL [45] | 7.08±1.84 | 17.46±1.51 | 3.83±1.46 | 9.26±1.19 | 4.13±1.69 | 7.35±1.38 | 54.86±1.59 | 62.71±1.17 | 5.54±0.98 | 7.06±0.85 | 7.13±1.04 | 7.70±0.64 |
| FOT [2] | 8.76±0.93 | 19.57±0.81 | 4.85±0.59 | 11.21±0.67 | 10.44±0.52 | 19.98±0.40 | 70.96±0.75 | 78.13±0.62 | 8.17±0.71 | 10.02±0.76 | 9.88±0.48 | 11.24±0.53 |
| FedWeIT [62] | 9.37±1.05 | 20.52±0.67 | 5.95±0.95 | 12.32±0.87 | 17.32±0.53 | 31.60±0.32 | 71.63±0.42 | 81.85±0.36 | 9.02±0.65 | 9.13±0.43 | 8.42±0.22 | 9.32±0.16 |

**With Sufficient Resources.** In Table 2, we test typical FCL methods under sufficient resource conditions. Notably, both FedCIL and AF-FCL methods excel, surpassing most established baselines on CIFAR-10 and Digit-10. This superior performance underscores the effectiveness of their generative models, which can effectively train on relatively simple datasets and generate high-quality samples for replay. However, as we delve into more complex and larger-scale datasets like CIFAR-100 and Tiny-ImageNet, we observe a significant drop in performance for both FedCIL and AF-FCL. This decline highlights the limitations of their generative models when faced with greater data diversity and complexity. In the context of federated domain-incremental learning, where the data distribution changes over time, GLFC demonstrates an interesting behavior. When there are no new incremental sample classes, it reverts to the FedAvg algorithm. On the other hand, Re-Fed consistently achieves reasonable performance across all datasets and scenarios. Its synergistic replay strategy against data heterogeneity seems to provide a robust foundation for FCL.

**With Limited Resources.** The experiments in Table 3 conducted under extremely limited resource conditions starkly contrast to the results obtained with sufficient resources. By simultaneously restricting three key types of training resources, we observe a significant decline in the performance of all methods. This underscores the critical importance of these resources for the effectiveness of federated continual learning techniques. To better understand the impact of these resource constraints, we conducted experiments from two perspectives. First, we analyzed how each resource limitation affects the performance of the various methods. This allowed us to identify the most sensitive components and bottlenecks that hinder the effectiveness of the FCL techniques under limited conditions. Second, we explored the interactions between different resource constraints and how they compound the challenges faced by the FCL methods. By understanding these interactions, we can gain insights into designing more resource-efficient FCL techniques. A detailed analysis will be provided in the next section.

In summary, the experiments with extremely limited resources highlight the need for continued research into developing FCL methods that can operate effectively under constrained conditions. By addressing the challenges of limited resources, we can pave the way for more practical and scalable applications of federated continual learning in real-world scenarios.

### 3.2.1 Ablation Study on Three Training Resources

We first explore the impact of each training resource on model performance and empirically select six baselines that may be sensitive to the corresponding resources and conduct experiments.

**1. Does the Memory Buffer matter?** As shown in Fig.1a, we select six methods that need extra memory buffer to cache previous samples/synthetic samples/auxiliary datasets. Specifically, when allocated a modest amount of memory buffer, these methods experience a marked improvement in their performance. This enhancement becomes particularly pronounced for methods that leverage generative models and distilled data for datasets with simple features. However, as the complexity of the datasets increases, posing challenges such as a wider diversity of classes and domains, the performance of these methods, particularly those reliant on generative modeling, undergoes a noticeable decline. This underscores the limitations of solely relying on synthetic data generation or distillation in complex learning environments. In contrast, methods that employ straightforward sample caching exhibit a more resilient performance profile. They maintain a relatively stable performance, demonstrating their robustness against dataset complexity. Nevertheless, as the number of cached samples surpasses a certain threshold, the model's performance gradually approaches an upper limit, indicating diminishing returns from further sample accumulation. *This plateauing effect underscores the need for a delicate balance in managing the memory buffer, aiming to maximize performance gains without incurring unnecessary computational cost.*

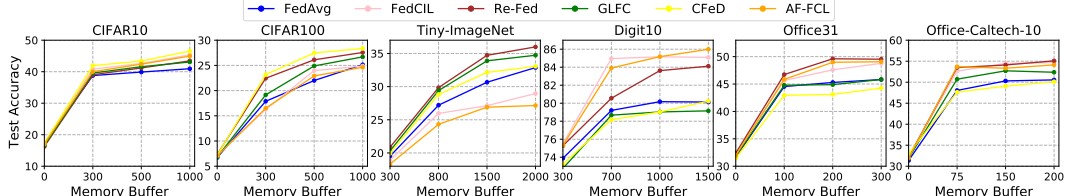

(a) The impact of the **Memory Buffer**. We set four different memory buffers for each dataset (e.g., $M = \{0, 300, 500, 1000\}$ for both CIFAR-10 and CIFAR-100). Here, we select six methods that use the memory buffer to cache samples from previous tasks, synthetic samples, or distillation samples.

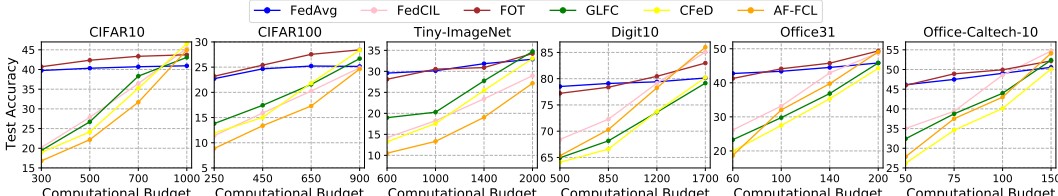

(b) The impact of the **Computational Budget**. Here we choose the FCL methods involved in complex computing. To normalize for effective computing due to the overhead of associated extra forward passes to decide on the distillation and training of the generative model, these additional calculations will share the computational budget equally with the target model.

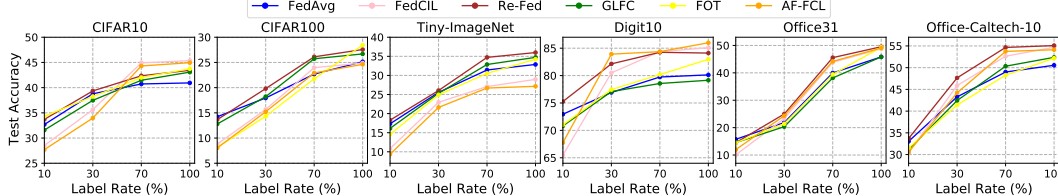

(c) The impact of the **Label Rate**. Here we equally test six random FCL methods with the ratio $R = \{10\%, 30\%, 70\%, 100\%\}$ for all datasets. Each FCL method trains the model just with the labeled data.

Figure 1: Ablation study on three training resources.

**2. Does the Computational Budget matter?** As depicted in Fig.1b, we carefully allocated varying computational budgets for each dataset, taking into account the maximum allowable limit specified in Table 5. It is evident from the results that when the computational budgets assigned to the various methods are restricted, the overall performance of the models inevitably suffers. This underlines the critical role that computational resources play in ensuring optimal model performance. However, certain methods demonstrate remarkable resilience, notably FedAvg and FOT. Even with limited computational budgets, they maintain satisfactory performance levels, highlighting their efficiency and effectiveness in resource-constrained environments. On the other hand, when the computational budget is abundant, allowing for unrestricted resource allocation, these other methods display the potential for higher performance ceilings. Given sufficient resources, they may achieve performance that surpasses that of FedAvg and FOT. *Nevertheless, the practical implications of such high resource requirements must be carefully considered, as they may not always be feasible or cost-effective in real-world applications.*

**3. Does the Label Rate matter?** In Fig.1c, we analyze the relationship between the label rate and the model performance. Our analysis reveals a compelling trend: when the label rate falls below 70%, a strategic increase in the proportion of labeled data within the dataset is a potent catalyst for enhancing the performance of most model approaches. However, the Digit-10 dataset stands out as a unique case. Despite boasting a vast quantity of data, the simplicity of its feature information, characterized by single-channel data, renders it less reliant on an extensive labeled dataset for optimal performance. Consequently, even with a relatively modest amount of labeled data, the model can achieve promising results, underscoring the importance of considering the dataset's inherent characteristics when evaluating the label rate's impact. This observation underscores the nuanced interplay between the label rate, dataset complexity, and model performance. *While a*

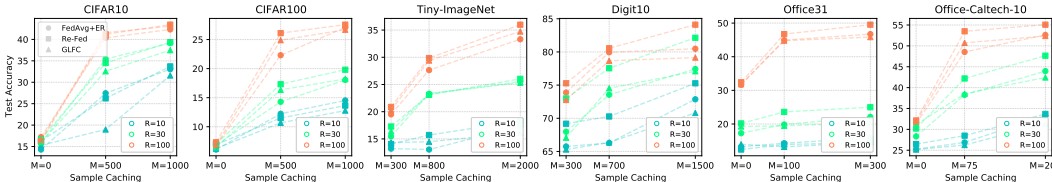

(a) **Sample Caching**. Here we select three FCL methods that require a cache of samples from previous tasks. We designed three combinations of memory buffers and label rates to conduct the experiments.

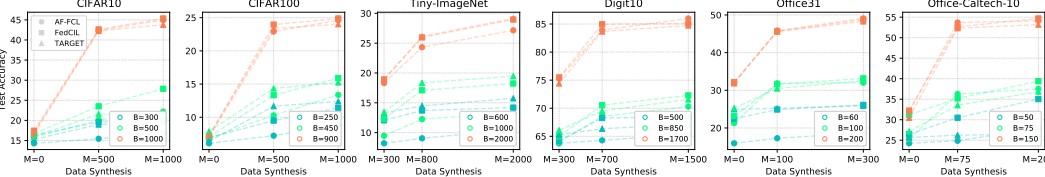

(b) **Data Synthesis**. Here we select three FCL methods that are involved in generative models. We designed three combinations of memory buffers and computational budgets to conduct the experiments.

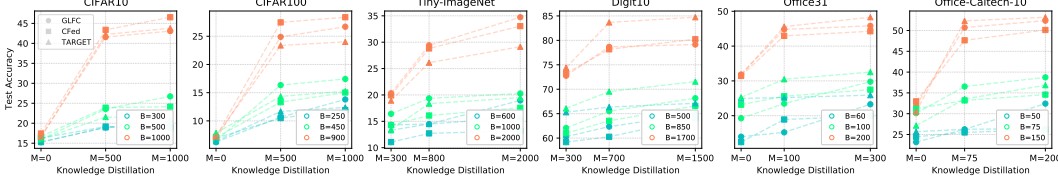

(c) **Knowledge Distillation**. Here we select three FCL methods that employ knowledge distillation. We designed three combinations of memory buffers and computational budgets to conduct the experiments.

Figure 2: Analysis of three typical FCL techniques.

*higher label rate generally leads to improved performance, the extent of this improvement can vary significantly depending on the dataset's characteristics and the model being employed.*

**Conclusion.** Existing methods have encountered limitations in achieving optimal results within resource-constrained environments. Notably, deploying a modestly sized memory buffer has emerged as a crucial factor in significantly augmenting model performance. This enhancement underscores the importance of an adequately provisioned yet economically efficient memory allocation strategy, as an overly generous buffer does not necessarily translate into commensurate gains in performance. A similar trend is observed with the label rate, emphasizing the need for a balanced approach in labeling data to maximize model efficacy. Moreover, the influence of the computational budget on FCL performance is intimately tied to the algorithmic design itself. Simple yet effective methods such as FedAvg demonstrate remarkable resilience, maintaining robust performance even with stringent computational constraints. This finding underscores the potential of streamlined FCL algorithms in addressing real-world challenges posed by limited resources. Experimental evidence highlights the imperative of selecting FCL methods tailored to the specific training resources. Researchers can achieve more effective learning outcomes by aligning algorithmic complexity with computational and memory resources.

### 3.2.2 Analysis of Typical FCL Techniques

In the experiments mentioned above, we have already observed that training resources play a crucial role in the performance of existing FCL methods. Next, we will further explore how to consider resource constraints in the design of FCL algorithms. We will research the four techniques above, often used to alleviate catastrophic forgetting, separately to demonstrate the effectiveness of different FCL techniques to different training resources.

**1. Does the Sample Caching matter?** To gain a deeper understanding of the efficacy of sample caching technology within limited resources, we impose restrictions on the memory buffer size and the label rate illustrated in Fig.2a. Regarding the sample caching technique, the label rate is pivotal in determining the quantity and quality of samples available for local training. Specifically, a higher label rate translates into more high-quality, labeled samples within the dataset, significantly enhancing the learning process. The Re-Fed algorithm, in particular, introduces an innovative approach that

leverages collaborative storage to intelligently select and retain the most critical samples when both the label rate and memory buffer are limited. This strategy is particularly effective in boosting model performance by focusing on the most informative data points, even under stringent resource constraints. Furthermore, as the label rate and memory buffer size vary, they jointly influence the performance of the sample caching technique in complex ways. The label rate not only dictates the overall quality of the data but also serves as an upper bound on the number of samples that can be feasibly stored within the given memory buffer. *Consequently, the effectiveness of the sample caching is intimately tied to the interplay between these two training resources. Optimizing the allocation and utilization of these resources is thus crucial for maximizing the benefits of sample caching with limited resources.*

**2. Does the Data Synthesis matter?** Data synthesis techniques, which often harness the power of generative models, are instrumental in creating synthetic data for replay purposes in various applications. In FCL, where resource constraints are prevalent, the training of these generative models inevitably introduces additional computational overhead. Consequently, it becomes imperative to carefully manage not only the memory buffer but also the computational budget overhead to ensure the feasibility and efficiency of the data synthesis process. As depicted in Fig.2b, the success of the data synthesis technique is intricately linked to both the available computational budget and the memory buffer capacity. Initially, as the memory buffer size increases, it provides more room for storing and manipulating synthetic data, thereby enhancing the technique's effectiveness. However, after reaching a threshold where a sufficient quantity of data can be comfortably accommodated, the marginal benefit of further expanding the memory buffer gradually diminishes. At this point, the computational budget becomes the primary bottleneck as the increased computational demands associated with generating higher-fidelity synthetic data or processing larger datasets begin to outweigh the benefits gained from additional memory. *This shift in the dominance of factors underscores the importance of striking a balance between memory and computational resources in the design and implementation of data synthesis techniques for FCL.* Efficient allocation of these resources can help maximize the effectiveness of the technique while minimizing the overall overhead, enabling more robust and scalable FCL systems.

**3. Does the Knowledge Distillation matter?** Knowledge distillation is a prevalent technique in mitigating catastrophic forgetting within existing FCL methods. While this approach effectively preserves key information across learning tasks, it necessitates the utilization of additional distillation datasets, whether publicly accessible or synthetically generated. Unfortunately, this requirement introduces additional complexity layers, including managing both computational overhead and memory buffer capacity. We primarily focus on limiting the computational budget, as it is often the primary factor constraining the scalability and speed of the distillation process. The Fig.2c illustrates that the computational budget is the primary limiting factor affecting the performance of the corresponding knowledge distillation methods. On the other hand, the memory buffer, though still a consideration, has a relatively minor impact on the overall performance of the knowledge distillation technique. This observation highlights the efficiency of the distillation process, where even a modest quantity of distilled data samples can achieve satisfactory performance gains. By efficiently managing the memory buffer to accommodate these necessary samples, we can maximize the benefits of knowledge distillation without incurring excessive memory overhead. *In summary, our approach to knowledge distillation in FCL involves a careful balancing act between computational budget and memory buffer, with a greater emphasis on optimizing the former to ensure the technique's practicality and performance within the confines of federated and continual learning environments.*

**Discussion about the Network Extension.** In this section, we do not perform additional experimental investigations specifically tailored for the Network Extension method. The main reason is that prevalent network extension methodologies typically presuppose knowledge of task boundaries, implying that the task ID for incoming data is accessible during inference. When this impractical assumption is relaxed, and we analogize to other methodologies that infer the category of the sample without prior task information, these extensions fail to uphold their erstwhile promising performance, as evidenced in Tables 2 and 3.

**Conclusion.** Three FCL strategies differ in strengths and limitations under varying resource conditions. With ample computational power and memory, sample caching preserves representative previous samples to combat forgetting, while data synthesis generates pseudo samples without storing real data – albeit requiring significant computational resources for effective rehearsal. Conversely,

knowledge distillation demonstrates resource efficiency by maintaining performance with minimal distilled samples, bypassing large memory buffers. Resource availability thus dictates strategy selection: sample caching and data synthesis excel in resource-rich environments, while knowledge distillation becomes optimal under constraints. Future research should prioritize reducing synthesis computation, optimizing cached memory usage, and enhancing distillation efficiency to address growing demands for resource-constrained FCL implementations.

# 4   Evaluation & Future Research Direction

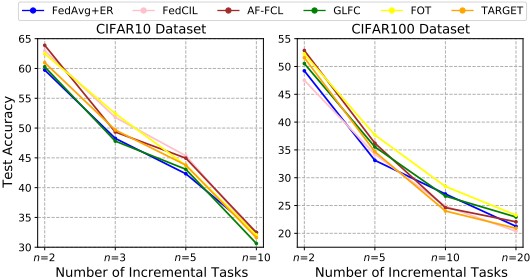

Figure 3: Performance w.r.t number of incremental tasks $n$ for class-incremental datasets.

Table 4: Performance comparison of various methods with a different order of domain tasks. We report the reverse order compared to the Table 2.

| Dataset | Metric | FedAvg | FedCIL | Re-Fed | FOT |
|---|---|---|---|---|---|
| Digit10 | $A(f)$ | 75.84±0.36 | 79.96±1.12 | 79.63±0.53 | 74.45±0.82 |
| | $\bar{A}$ | 83.62±0.41 | 87.98±0.88 | 88.23±0.49 | 83.91±0.76 |
| Office31 | $A(f)$ | 36.65±0.52 | 41.90±1.36 | 42.71±0.44 | 40.29±0.94 |
| | $\bar{A}$ | 41.47±0.49 | 44.18±0.95 | 45.05±0.39 | 43.86±0.73 |

In this section, we delve deeper into the analysis of existing FCL methodologies by conducting extensive quantitative experiments that focus on two crucial aspects: the number of tasks within Class-IL datasets in Figure. 3 and the order of tasks across various Domain-IL datasets in Table 4. By systematically varying these parameters, we aim to gain a nuanced understanding of how these factors impact the performance of FCL methods. Firstly, we observe that as the number of tasks in the Class-IL datasets escalates, the performance of all the evaluated methods noticeably declines. Nevertheless, despite this performance decrement, the general trends exhibited by the different methods remain comparable, suggesting that the fundamental challenges they face are similar. Furthermore, our experiments reveal that the order in which domain tasks are presented to the model can significantly influence its ultimate performance within Domain-IL scenarios. This finding emphasizes the importance of task scheduling and curriculum design in FCL systems, as certain sequences may be more conducive to learning than others. However, even with this variability in performance due to task order, each method maintains its distinctive strengths and tendencies, demonstrating its resilience and adaptability to different contexts.

Moreover, although there is a lot of research on FCL, these studies mainly focus on transferring CL algorithms in centralized settings to alleviate catastrophic forgetting, ignoring the issue of training resource overhead. Based on this, we shed light on the future research direction in Appendix C.

# 5   Conclusion

This paper has presented a tutorial on FCL under resource constraints. Firstly, we begin with an introduction to the existing FCL methods and three training resources on distributed devices that influence the performance of the methods. Then, we conduct experiments using ablation studies on each training resource and analyze the effect of four typical FCL techniques on alleviating catastrophic forgetting in FCL. Finally, we discuss future research directions in resource-constrained FCL.

## Acknowledgments and Disclosure of Funding

This work is supported by the National Key Research and Development Program of China under grant 2024YFC3307900; the National Natural Science Foundation of China under grants 62376103, 62302184, 62436003 and 62206102; Major Science and Technology Project of Hubei Province under grant 2024BAA008; Hubei Science and Technology Talent Service Project under grant 2024DJC078; Ant Group through CCF-Ant Research Fund; and Fundamental Research Funds for the Central

Universities under grant YCJJ20252319. The computation is completed in the HPC Platform of Huazhong University of Science and Technology.

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

# A    Background and Related Work

**Federated Learning.** Federated Learning (FL) is an approach to developing a unified global model by combining models trained on local, private datasets from multiple clients [31, 32, 56, 39]. A notable FL framework, FedAvg [38], enhances the global model by averaging the parameters of these locally trained models. Nevertheless, conventional FL methods, such as FedAvg, encounter difficulties when dealing with data heterogeneity, where client datasets are not independently and identically distributed (Non-IID), leading to reduced model performance [13, 12, 11, 46]. To address the Non-IID problem in FL, an optimization technique incorporating a proximal term was introduced in [22] to alleviate the impact of diverse and Non-IID data distributions among devices. Another strategy, federated distillation [67, 72], focuses on transferring knowledge from multiple local models to the global model by aggregating solely the soft predictions produced by each model. The authors of [33] presented a knowledge distillation technique that employs unlabeled training data as a surrogate dataset. However, these methods are tailored to address static data with spatial heterogeneity and do not account for the challenges presented by temporally heterogeneous streaming tasks in FL.

**Continual Learning** Continual Learning (CL) is a machine learning paradigm that enables models to learn sequentially from a stream of tasks while preserving knowledge acquired from earlier tasks [51, 73, 70]. CL methods can be broadly categorized into three primary approaches: replay-based [47, 8], regularization-based [16, 17], and parameter isolation techniques [52, 43]. Replay-based methods involve retaining a subset of previous task samples to maintain knowledge when learning new tasks. Regularization-based methods prevent the overwriting of existing knowledge by applying constraints to the loss function during the learning of new tasks. Parameter isolation methods usually introduce extra parameters and computational resources to facilitate the learning of new tasks. In this work, we concentrate on federated continual learning, which integrates elements of both federated and continual learning.

**Federated Continual Learning** Federated Continual Learning (FCL) is introduced to tackle the challenge of learning from a sequence of tasks on each client, focusing on adapting the global model to new data while preserving knowledge from previous data [57, 55, 63]. Although important, FCL has only recently started to receive attention, with [62] serving as a seminal work in this area. This study concentrates on Task-Incremental Learning (Task-IL), which necessitates distinct task IDs at inference time and employs dedicated task-specific masks to improve personalized performance. Concurrently, research efforts like [64] consider the multi-granularity representation of knowledge, fostering the integration of spatial-temporal knowledge in Federated Continual Incremental Learning (FCIL). The authors of [30] align local domain knowledge with a dynamic network to strike a balance between resource usage and model performance. Other works, such as [37], employ a proxy dataset and leverage knowledge distillation at both the server and client levels. In contrast, [5, 4] simplifies the problem by assuming clients have sufficient storage to store and share old examples, deviating from the conventional FL framework. Additionally, studies like [6, 15] investigate FCL in applications beyond image classification.

# B    Problem Formulation

**Federated Learning.** In the context of Federated Learning (FL), a common problem involves collaboratively training a global model across $K$ clients. Each client $k$ has exclusive access to its local private dataset, denoted as $\mathcal{T}_k = \{x_k^{(i)}, y_k^{(i)}\}$, where $x_k^{(i)}$ represents the $i$-th data sample, and $y_k^{(i)} \in \{1, 2, \ldots, C\}$ is the associated label for $C$ classes. The size of dataset $\mathcal{T}_k$ is indicated by $|\mathcal{T}_k|$. The global dataset is the aggregate of all local datasets, i.e., $\mathcal{T} = \{\mathcal{T}_1, \mathcal{T}_2, \ldots, \mathcal{T}_K\}$, with $\mathcal{T} = \sum_{k=1}^{K} \mathcal{T}_k$. The goal of the FL system is to train a global model $w$ that minimizes the overall empirical loss across the entire dataset $D$:

$$\min_{w} \mathcal{L}(w) := \sum_{k=1}^{K} \frac{|\mathcal{T}_k|}{|\mathcal{T}|} \mathcal{L}_k(w),$$

$$\text{where } \mathcal{L}_k(w) = \frac{1}{|\mathcal{T}_k|} \sum_{i=1}^{|\mathcal{T}_k|} \mathcal{L}_{CE}(w; x_i^k, y_i^k). \tag{1}$$

Table 5: Experimental Details. Analysis of various considered settings of different datasets in the experiments section.

| Attributes | CIFAR-10 | CIFAR-100 | Tiny-ImageNet | Digit-10 | Office-31 | Office-Caltech-10 |
|---|---|---|---|---|---|---|
| Task size | 178MB | 178MB | 435MB | 480M | 88M | 58M |
| Image number | 60K | 60K | 120K | 110K | 4.6k | 2.5k |
| Image Size | $3\times32\times32$ | $3\times32\times32$ | $3\times64\times64$ | $1\times28\times28$ | $3\times300\times300$ | $3\times300\times300$ |
| Task number | $n = 5$ | $n = 10$ | $n = 10$ | $n = 4$ | $n = 3$ | $n = 4$ |
| Task Scenario | Class-IL | Class-IL | Class-IL | Domain-IL | Domain-IL | Domain-IL |
| Batch Size | $s = 64$ | $s = 64$ | $s = 128$ | $s = 64$ | $s = 32$ | $s = 32$ |
| ACC metrics | Top-1 | Top-1 | Top-10 | Top-1 | Top-1 | Top-1 |
| Learning Rate | $l = 0.01$ | $l = 0.01$ | $l = 0.001$ | $l = 0.001$ | $l = 0.01$ | $l = 0.01$ |
| Data heterogeneity | $\alpha = 1.0$ | $\alpha = 1.0$ | $\alpha = 10.0$ | $\alpha = 0.1$ | $\alpha = 1.0$ | $\alpha = 1.0$ |
| Client numbers | $C = 20$ | $C=20$ | $C=20$ | $C=15$ | $C=10$ | $C=8$ |
| Local training epoch | $E = 20$ | $E = 20$ | $E = 20$ | $E = 20$ | $E = 20$ | $E = 15$ |
| Client selection ratio | $k = 0.4$ | $k = 0.4$ | $k = 0.5$ | $k = 0.4$ | $k = 0.4$ | $k = 0.5$ |
| Communication Round | $T = 80$ | $T = 80$ | $T = 100$ | $T = 60$ | $T = 60$ | $T = 40$ |
| Memory Buffer | $M = 1000$ | $M = 1000$ | $M = 2000$ | $M = 1500$ | $M = 300$ | $M = 200$ |
| Label Rate | $R = 100\%$ | $R = 100\%$ | $R = 100\%$ | $R = 100\%$ | $R = 100\%$ | $R = 100\%$ |
| Computational Budget | $B = 1000$ | $B = 900$ | $B = 2000$ | $B = 1700$ | $B = 200$ | $B = 150$ |

Here $\mathcal{L}_k(w)$ signifies the local loss for client $k$, and $\mathcal{L}_{CE}$ is the cross-entropy loss function, which quantifies the discrepancy between the predicted and actual labels.

**Continual Learning.** In a typical Continual Learning (CL) scenario (outside of a federated context), a model is trained on a sequence of tasks $\{\mathcal{T}^1, \mathcal{T}^2, \ldots, \mathcal{T}^n\}$, where $\mathcal{T}^t$ represents the $t$-th task. Each task $\mathcal{T}^t = \sum_{i=1}^{N^t}(x_t^{(i)}, y_t^{(i)})$ comprises $N^t$ pairs of data samples $x_t^{(i)} \in \mathcal{X}^t$ and their corresponding labels $y_t^{(i)} \in \mathcal{Y}^t$. The domain space and label space for the $t$-th task are denoted by $\mathcal{X}^t$ and $\mathcal{Y}^t$, respectively, with $|\mathcal{Y}^t|$ classes. The total class set across all tasks is $\mathcal{Y} = \bigcup_{t=1}^n \mathcal{Y}^t$, and similarly, the total domain space is $\mathcal{X} = \bigcup_{t=1}^n \mathcal{X}^t$. This paper focuses on two CL scenarios: (1) *Class-Incremental Task*: All tasks share the same domain space ($\mathcal{X}^1 = \mathcal{X}^t$ for all $t \in [n]$), but the number of classes may vary ($\mathcal{Y}^1 \neq \mathcal{Y}^t$ for all $t \in [n]$). (2) *Domain-Incremental Task*: All tasks have the same number of classes ($\mathcal{Y}^1 = \mathcal{Y}^t$ for all $t \in [n]$), but the domain and data distribution change over tasks ($\mathcal{X}^1 \neq \mathcal{X}^t$ for all $t \in [n]$).

**Federated Continual Learning.** We combine the CL with the federated setting. Our objective is to train a global model for $K$ clients, where each client $k$ can only access its local sequence of tasks $\{\mathcal{T}_k^1, \mathcal{T}_k^2, \ldots, \mathcal{T}_k^n\}$. When the $t$-th task arrives, the goal is to train a global model $w^t$ across all $t$ tasks, denoted by $\mathcal{T}^t = \{\sum_{n=1}^t \sum_{k=1}^K \mathcal{T}_k^n\}$. This can be formulated as:

$$w^t = \arg\min_{w \in \mathbb{R}^d} \sum_{n=1}^t \sum_{k=1}^K \sum_{i=1}^{N_k^n} \frac{1}{|\mathcal{T}^t|} \mathcal{L}_{CE}(w; x_{k,n}^{(i)}, y_{k,n}^{(i)}). \tag{2}$$

Here $\mathcal{L}_{CE}$ is the cross-entropy loss function used to measure the discrepancy between predictions and true labels.

## B.1 Experiment Setup

### B.1.1 Datasets.

Our experiments utilize diverse datasets distributed across two federated incremental scenarios, encompassing six datasets in total.

**Class-Incremental Task Dataset:** This dataset gradually introduces new classes. It begins with a subset of classes and expands by adding more classes in subsequent stages, facilitating models to learn and accommodate an increasing range of classes.

*(1) CIFAR-10 [18]:* Comprises 10 object classes, encompassing everyday items, animals, and vehicles. It includes 50,000 training and 10,000 test images.

*(2) CIFAR-100 [18]:* Similar to CIFAR-10 but with 100 detailed object classes. It contains 50,000 training and 10,000 test images.

*(3) Tiny-ImageNet [20]:* A selection from the ImageNet dataset, featuring 200 object classes. It includes 100,000 training images, 10,000 validation images, and 10,000 test images.

**Domain-Incremental Task Dataset:** This dataset progressively introduces new domains. It starts with samples from a specific domain and incorporates additional domains in later stages, allowing models to generalize to new contexts.

*(1) Digit-10:* The Digit-10 dataset includes 10 digit categories across four domains: MNIST [21], EMNIST [3], USPS [14], and SVHN [40]. Each dataset represents a specific domain, such as handwriting style, and contains 10 digit classes.

- MNIST: Contains 60,000 training and 10,000 test handwritten digit images.
- EMNIST: An extension of MNIST with 240,000 training and 40,000 test handwritten character (letter and digit) images.
- USPS: Includes 7,291 training and 2,007 test handwritten digit images from the US Postal Service.
- SVHN: Features 73,257 training and 26,032 test images of house numbers captured from Google Street View.

*(2) Office-31 [50]:* Contains images from three domains: Amazon, Webcam, and DSLR. It covers 31 categories, with approximately 4,100 images per domain.

*(3) Office-Caltech-10 [69]:* Includes images from four domains: Amazon, Caltech, Webcam, and DSLR. It comprises 10 object categories, with around 2,500 images per domain.

### B.1.2 Baselines.

In this paper, we follow the same protocols proposed by [38, 47] to set up tasks. We evaluate our resource-constrained settings with the following baselines.
**FedAvg** [38]: It is a representative FL model that aggregates client parameters in each communication round. It is a simple but effective model for FL.

**FedProx** [22]: It is also a representative FL model, which is better at handling heterogeneity in federated networks than FedAvg.

**FedAvg/FedProx+ER [49]:** Experience Replay is a technique primarily utilized in reinforcement learning, especially in deep reinforcement learning algorithms. Here, we combine the ER algorithm with the FL algorithms FedAvg and FedProx.

**GLFC** [5]: This method addresses the federated class-incremental learning issue and trains a global model with additional class-imbalance losses. A proxy server is used to reconstruct samples to help select the best old models for model updates.

**FedCIL** [45]: This method employs the ACGAN network to generate synthetic samples to consolidate the global model and align sample features in the output layer. The authors conducted experiments in the FCIL scenario, and we adopted it in our FDIL setting.

**TARGET** [66]: This method uses a generator to synthesize data to simulate the global data distribution on each client without additional data from previous tasks. The data heterogeneity is considered with catastrophic forgetting, and the authors focus on the FCIL scenario.

**AF-FCL** [58]: This method focuses on the data noise among previous tasks, which degrades the model performance. The NF model is used to quantify the credibility of previous knowledge and select the transfer knowledge.

**Re-Fed** [23]: This approach proposes synergistic replay for each client to selectively cache samples based on the understanding of both local and global data distributions. Each client trains an additional personalized model to discern the importance of the score to each client.

**FOT** [2]: This method uses orthogonal projection technology to project different parameters into different spaces to isolate task parameters. In addition, this method proposes a secure parameter

aggregation method based on projection. In the model inference stage, the method requires assuming that the boundaries of the task are known. We modified it to an automated inference of the model.

**CFeD** [36]: It employs public datasets as proxy data for knowledge distillation at both client and server sides. Upon introducing new tasks, clients use proxy data to transfer knowledge from previous to current models thus mitigating inter-task forgetting.

**FedWeIT** [62]: This method divides parameters into task-specific and shared parameters. A multi-head model based on known task boundaries is used in the original method. To ensure a fair comparison, we modify it to automate the inference of the model.

## C   Future Research in Resource-Constrained Federated Continual Learning

In this paper, based on extensive experiments, we have proved that training resources play a crucial role in the effectiveness of existing FCL methods. Therefore, we propose the following suggestions for future research:

- *Sampling Strategy:* Sample caching is a mainstream method to prevent catastrophic forgetting in FCL, and it often has good results when memory buffer resources are not tight. In our experiments, we used simple strategies such as Random, FIFO, and LIFO. However, apart from the collaborative replay strategy proposed in [23], there is currently no in-depth exploration of sample caching strategies in FCL. In addition, future research can also focus on directly selecting exemplar samples during local training to train and cache locally.

- *Update Optimization:* Common SGD optimization often achieves expected performance under IID data, but in FCL, data exhibits heterogeneity in both spatial and temporal dimensions. Better optimization approaches can be considered, such as using analytic learning to estimate parameters and employing sharpness-aware minimization to replace empirical risk minimization), ultimately enhancing the model's robustness to data heterogeneity and improving model performance.

- *Bandwidth Restriction:* Bandwidth plays a crucial role in distributed systems, as the communication frequency between different clients and servers directly impacts the convergence of the model. In FCL, due to the continuous arrival of incremental tasks, clients need to participate in federated training for extended periods, resulting in substantial communication overhead. Researching how to reduce this communication overhead is of great significance for the large-scale deployment of FCL.

- *Adaptive Allocation:* Due to limited training resources, optimizing their allocation to boost model performance represents a pragmatic approach. This could entail allocating appropriate memory buffers to different datasets and assigning computational resources to particular processes. One viable method is the application of combinatorial optimization techniques. In practical implementations or method designs, dynamically allocating resources to each technique based on availability can further improve performance.

