# OpenReview forum: "Resource-Constrained Federated Continual Learning: What Does Matter?"
_NeurIPS.cc/2025/Conference — NeurIPS 2025 poster_

### Official Review · Reviewer_SBXG · 2025-07-02

**Clarity:** 4
**Significance:** 3
**Originality:** 4
**Rating:** 5
**Confidence:** 5

**Summary:**

This paper focuses on Federated Continual Learning (FCL) in resource-constrained scenarios. A large-scale benchmark involving six datasets and two incremental learning scenarios is constructed. Experimental results reveal that resource constraints substantially degrade the performance of existing FCL approaches. Based on experimental results, the paper summarized practical insights and future directions.

**Questions:**

1. The LLM foundation is a hot topic. Is there LLM based FCL methods? Are these methods applicable to the resource-constrained setting mentioned in this paper?
2. This empirical study focuses on the image classification task. Do existing FCL methods work on NLP tasks?

**Ethical Concerns:**

["NO or VERY MINOR ethics concerns only"]

**Final Justification:**

After reading the rebuttal, I decide to keep my score.

**Limitations:**

yes

**Quality:**

3

**Strengths And Weaknesses:**

Strengths:
1. The paper is well-organized and the research topic is hot.
2. It is timely and meaningful to study the resource constraints that affect the deployment of FCL.
3. Extensive experiments and analysis are provided with significant workloads.
4. The relative literatures are summarized well.
5. This work may shed light on the future research directions for resource-friendly FCL.

Weaknesses:
1. Communication overhead is a key resource constraint in FL. The paper focuses on three specific resources. The authors should justify why other constraints, particularly communication overhead, were not selected for analysis.
2. In Figure 1, some methods appear to underperform FedAvg even with sufficient resources, contradicting findings in their original papers. The authors should explain this discrepancy.
3. The authors measure computational cost using gradient steps. As training time is a common alternative metric in FL, the authors should clarify their choice of gradient steps.

---

> ### Author Rebuttal · Authors · 2025-07-29
>
> > **W1. The authors should justify why other constraints, particularly communication overhead, were not selected for analysis.**
>
> **R1:** Thank you for your insightful comments. We fully agree that communication overhead is one of the core challenges in FL, especially in real-world deployments. In this work, we did not explicitly include communication cost as one of the three core resource constraints, mainly due to the following considerations:
>
> **Communication cost is fundamentally different from the client-side training constraints we focus on.** Our study primarily targets three types of **client-local resource limitations**: memory buffer, computational budget, and label rate. These are hard constraints that directly affect the training capacity of edge devices. In contrast, communication cost is mainly determined by factors such as communication frequency, model synchronization strategy (e.g., number of rounds $T$), and model size. **Therefore, it impacts training latency rather than the local training capability of clients.**
>
> **Communication constraints deserve dedicated treatment in future studies.** We appreciate your point that communication cost is an important dimension in FCL. For example, future studies could explore communication heterogeneity (e.g., bandwidth and latency across clients), the applicability of compression/sparsification techniques in FCL, or trade-offs between communication and local computation. As noted in Appendix E, we have already listed communication constraints as a direction for future work. In the current paper, we aim to first provide a systematic understanding of client-side resource limitations, which we believe is a necessary step toward building a more comprehensive performance analysis framework for FCL.
>
> > **W2. In Figure 1, some methods appear to underperform FedAvg even with sufficient resources, contradicting findings in their original papers.**
>
> **R2:** We thank the reviewer for the careful observation regarding the performance differences in the “resource-sufficient” setting. We understand the concern that some methods perform worse than FedAvg even when resources are sufficient, and we offer the following clarifications:
>
> 1. **Inherent differences due to scenario adaptation**
>
>    Some methods (e.g., FOT and FedWeIT) were originally designed for specific settings, such as pure Class-IL with known task boundaries. In our benchmark, we enforce a unified protocol across both Class-IL and Domain-IL settings **without assuming task boundaries or task IDs**, which requires adapting these methods beyond their original design assumptions. This generalization may cause performance degradation on certain datasets.
>
> 2. **Differences in the definition of “resource sufficiency”**
>
>    In prior work, “sufficient resources” often refers to a single unconstrained dimension (e.g., unlimited memory). In contrast, our benchmark defines the **resource-sufficient setting** as the baseline configuration where **memory buffer (M), computation budget (B), and label rate (R)** are all reasonably provisioned (see Table 4). For instance, generative replay methods in prior work often assume unlimited training iterations for generators. However, in our setup, generator training shares a fixed computational budget with the target model, potentially reducing sample quality and thus final performance.
>
> > **W3. The authors measure computational cost using gradient steps. As training time is a common alternative metric in FL, the authors should clarify their choice of gradient steps.**
>
> **R3:** Thank you for the question. We chose to use the number of gradient steps (i.e., the number of `backward()` calls) rather than wall-clock training time as a metric for quantifying computation cost, primarily to ensure experimental fairness. The reasons are as follows:
>
> **Gradient steps serve as a hardware-independent, architecture-agnostic, and more controllable indicator of computation cost.** Compared to wall-clock time, gradient steps offer several advantages:
>
> - **They are independent of hardware configuration** (e.g., GPU/CPU performance, memory size), which avoids variability in computation overhead caused by differences in deployment environments across methods;
> - **They are less sensitive to implementation details** such as the optimizer used or data loading mechanisms, making them more stable and reliable for cross-method comparisons;
> - **They provide direct controllability and reproducibility**: by specifying the maximum number of `backward()` calls, we can precisely bound the amount of client-side computation per communication round.
>
> > **Q1. The LLM foundation is a hot topic. Is there LLM based FCL methods? Are these methods applicable to the resource-constrained setting mentioned in this paper?**
>
> **A1:** We appreciate the reviewer’s insightful question regarding the integration of LLMs into FCL. LLMs have gained significant attention due to their strong performance on a wide range of tasks. However, their **substantial computational and communication overhead** poses major challenges for deployment on edge devices. Incorporating LLMs into resource-constrained FCL settings would require **a fundamental rethinking of model compression, distributed optimization, and continual learning strategies**.
>
> Currently, most FCL methods focus on traditional architectures (e.g., CNNs, ResNets), which are more compatible with edge environments. We acknowledge that adapting LLMs to FCL under resource constraints is an important and promising direction. We plan to explore this line of work in future research, building on the foundation established in this study.
>
> > **Q2. This empirical study focuses on the image classification task. Do existing FCL methods work on NLP tasks?**
>
> **A2:** We thank the reviewer for raising the important question regarding natural language processing (NLP) tasks. To date, the vast majority of existing FCL methods have been proposed and evaluated in the context of image classification, while **systematic studies in NLP scenarios remain largely lacking**. Image classification tasks offer well-standardized data structures and widely used datasets, making it feasible to impose consistent resource constraints (M/B/R) and perform fair comparisons across methods. To ensure **comparability and reproducibility**, we selected image classification as the experimental foundation in this work. We agree that extending FCL benchmarks to NLP is of great value, and we plan to **systematically explore the design and evaluation of FCL under NLP scenarios** in future work, thereby enhancing the general applicability and practical relevance of our study.

---

> > ### Comment · Reviewer_SBXG · 2025-08-02
> >
> > Thanks for the detailed rebuttal. All of my concerns have been addressed.

---

### Official Review · Reviewer_duEh · 2025-07-02

**Clarity:** 2
**Significance:** 3
**Originality:** 3
**Rating:** 4
**Confidence:** 4

**Summary:**

The paper benchmarks state-of-the-art approaches applied to Continual Learning in the context of federated learning in resource-constrained settings. From their experiments, they conclude that all existing Federated Continual Learning approaches fail to achieve expected performance in resource-constrained settings - Memory Buffer, Computational Budget, and Label Rate.

**Questions:**

The one key contribution stated by the authors is the future research directions based on their benchmarking experiments, which the authors ask us to read from Appendix E. To be fair, we can evaluate the merits of the paper only by looking at the main contents of the paper. The authors should consider reorganizing the contents in a way that key contributions and takeaways stay in the main pages of the paper, and only supplementary information should be moved into the Appendix.

**Ethical Concerns:**

["NO or VERY MINOR ethics concerns only"]

**Final Justification:**

The response from the authors is convincing and satisfactory.

**Limitations:**

Yes

**Quality:**

2

**Strengths And Weaknesses:**

Overall, it is an interesting research problem to understand the impact of resource constraints in Federated Continual Learning.

However, the paper has some weaknesses that stem from the paper's organization and writing clarity, which hinders its effectiveness.

1. Lack of clarity in experimental takeaways- key takeaways from experiments (3.2) before the ablation study are not clear. Since this is a benchmarking paper, it is really important to clearly highlight the experimental takeaway and conclusions, and subsequent discussions.

2. Insufficient explanation of key concepts -  For Example, Section 3.1.1. It would be good to explain in one or two sentences what Class Incremental learning and Domain Incremental Learning are. The paper should be self-contained, rather than always asking the readers to refer to the Appendix. It's important to explain these concepts clearly as they set the premise of the research problem discussed in the paper.

3. Missing citations in introduction from lines 55 to 85; it's important to cite when you mention other works. For example, in lines 55 and 56, for many existing methods, particularly those based on replay [cite], additionally, some other methods relying on regularization [cite]

4. Authors should stay within the allowed page limit to explain their work. The appendix should be used only for supplementary information, not for discussing any of the key contributions of the paper.

---

> ### Author Rebuttal · Authors · 2025-07-29
>
> > **W1. Key takeaways from experiments (3.2) before the ablation study are not clear.**
>
> **R1:** We sincerely thank the reviewer for the insightful feedback. Below, we clarify the design rationale of Section 3.2 and summarize the key findings to address your concerns.
>
> The structure of **Section 3.2** is intended to reflect a progressive analysis — from holistic observation to root cause attribution. **We first establish baseline comparisons by contrasting “resource-rich” and “resource-constrained” settings**, allowing us to assess the inherent strengths and weaknesses of various method families (e.g., generative replay, sample-based replay, knowledge distillation). In resource-rich scenarios, we observe, for example, the limitations of generative models on complex datasets, which serves as a reference point for later analyses under constrained conditions.
>
> Subsequently, through a structured ablation study, we control combinations of memory buffer (M), computational budget (B), and label rate (R), aiming to **isolate and analyze the independent and joint effects of different resource dimensions on performance**. This design allows us to identify specific bottlenecks for different methods — for instance, the strong dependency of generative models on compute, or the sensitivity of replay-based methods to memory buffer size — providing empirical evidence to support the core conclusions.
>
> To better emphasize the role of ablation studies in Section 3.2, we will revise the main text to explicitly highlight the following findings:
>
> - **Method performance is strongly tied to dataset complexity.** Generative replay methods perform well on simple datasets but degrade significantly on more complex ones, revealing their limited adaptability to high-diversity data.
> - **Auxiliary components (e.g., generators, distillation modules) amplify the impact of resource constraints.** When all three critical resources (memory, computation, labeling) are simultaneously constrained, all methods experience significant performance drops. In contrast, base methods that do not rely on additional modules (e.g., FedAvg + FedProx) demonstrate the most robust behavior.
>
> > **W2.Insufficient explanation of key concepts.**
>
> **R2:** We thank the reviewer for the valuable suggestion regarding the clarity of key concepts. In response, we will revise Section 3.1.1 to explicitly include the definitions of **Class Incremental Learning (Class-IL)** and **Domain Incremental Learning (Domain-IL)** in the main text, rather than placing them in the appendix. This adjustment aims to help readers quickly grasp the core task settings used throughout our experiments.
>
> - **Class-Incremental Learning:** A learning scenario where models sequentially learn tasks with shared data domains but new classes in each task, requiring adaptation to new classes without forgetting old ones.
> - **Domain-Incremental Learning:** A learning scenario where tasks involve fixed classes but varying data domains (e.g., different environments), demanding adaptation to domain shifts while maintaining recognition of all classes.
>
> > **W3. Missing citations in introduction from lines 55 to 85.**
>
> **R3:** Thank you for your detailed feedback. We fully agree that explicitly citing related work in the introduction is essential for academic rigor. In response to your comment regarding the paragraph discussing replay and regularization-based methods, we have revised the text to include specific citations based on the references already used in our paper. The updated version is as follows:
>
> “Memory buffer serves as a crucial training resource for many existing methods, particularly those based on replay [24, 41], as they necessitate the storage of previous samples or synthetic data for retraining. Additionally, some other methods relying on regularization [1, 34] also require buffering a portion of sample data or utilizing extra auxiliary datasets to facilitate the training process.”
>
> [1] Sara Babakniya, Zalan Fabian, Chaoyang He, Mahdi Soltanolkotabi, and Salman Avestimehr. A data-free approach to mitigate catastrophic forgetting in federated class incremental learning for vision tasks. Advances in Neural Information Processing Systems, 36, 2024.
>
> [24] Yichen Li, Qunwei Li, Haozhao Wang, Ruixuan Li, Wenliang Zhong, and Guannan Zhang. Towards efficient replay in federated incremental learning. The Thirty-Fifth IEEE/CVF Conference on Computer Vision and Pattern Recognition (CVPR 2024), Seattle, USA, June 17-21, 2024.
>
> [34] Yuhang Ma, Zhongle Xie, Jue Wang, Ke Chen, Lidan Shou, and Luc De Raedt. Continual federated learning based on knowledge distillation. In IJCAI, pages 2182–2188, 2022.
>
> [41] Daiqing Qi, Handong Zhao, and Sheng Li. Better generative replay for continual federated learning. arXiv preprint arXiv:2302.13001, 2023.
>
> > **W4&Q1. Concerns about page limits and appendix usage.**
>
> **R4:** We thank the reviewer for highlighting the importance of structural clarity. We fully understand the conference's page limit requirements and agree that key content should be prioritized in the main text to ensure that readers can follow the paper's core contributions without needing to frequently consult the appendix. In response, **we will reorganize the division between the main text and the appendix**, ensuring that essential information is properly integrated into the main body, while strictly adhering to the length constraints. This will help improve the overall clarity and presentation quality of the paper.

---

### Official Review · Reviewer_ef6U · 2025-07-03

**Clarity:** 3
**Significance:** 2
**Originality:** 3
**Rating:** 3
**Confidence:** 4

**Summary:**

This paper presents an extensive empirical investigation into Federated Continual Learning (FCL) methods under practical resource limitations, focusing on memory constraints, computational budgets, and varying label availability. The authors evaluate over 10 FCL algorithms across six datasets, examining both Class-Incremental and Domain-Incremental learning scenarios. Their rigorous experimentation (exceeding 1,000 GPU hours) reveals a concerning performance gap: most current state-of-the-art FCL approaches significantly degrade when operating within realistic resource boundaries. The work systematically analyzes four fundamental FCL techniques—sample caching, data synthesis, knowledge distillation, and network extension—providing valuable insights into their relative effectiveness under various constraint profiles.

**Questions:**

1）：Based on the existing experimental findings and insights, developing a novel method that effectively performs FCL under resource-constrained conditions would significantly enhance the paper's contribution to the community. The current analysis provides valuable empirical observations that could serve as a foundation for methodological innovation.

2）：The experimental setup presents idealized resource constraints that may not fully reflect real-world deployment challenges. Further analyzing memory and computation in terms of MB and FLOPs would provide more practical insights when deploying to mobile devices. Additionally, resources in real-world scenarios often fluctuate dynamically - a critical aspect overlooked by the current experimental framework. The paper should explore whether similar insights hold under dynamically changing resource conditions to better reflect practical applications.

3）：Regarding dataset selection, testing on a broader range of datasets would better evaluate the effectiveness of these methods while validating the robustness of the insights. The current focus on CIFAR and its variants may not capture the complexity and diversity of real-world data. Expanding experiments to include more challenging datasets such as text, speech, and multimodal data would strengthen the paper's claims and increase its practical relevance.

**Ethical Concerns:**

["Major Concern: Data privacy, copyright, and consent", "Major Concern: Safety and security"]

**Final Justification:**

I still think this paper is not good enough to be accepted.

**Limitations:**

Yes

**Paper Formatting Concerns:**

No formatting issues were identified in the paper. The manuscript appears to adhere to the required formatting guidelines with appropriate structure, citation style, and layout.

**Quality:**

3

**Strengths And Weaknesses:**

**Strength**

1）：The paper addresses a critical yet underexplored gap—real-world deployment feasibility of FCL systems under tight resource budgets (memory buffer, computational budget, and label rate data), which is vital for edge computing and mobile devices.

2）： Extensive experiments (totaling over 1,000 GPU hours) across six datasets and more than ten baseline methods evaluate the performance of various approaches under different resource-constrained settings.

3）：This paper provides valuable insights to guide future FCL method design. The three FCL strategies exhibit distinct strengths and limitations under varying resource conditions. For instance, knowledge distillation demonstrates remarkable resource efficiency by maintaining performance with minimal distilled samples, effectively circumventing the need for large memory buffers.



**Weaknesses**：

1）：The paper primarily analyzes existing methods without proposing a novel approach to address the identified problems, limiting its contribution to empirical insights. This lack of methodological innovation diminishes its novelty and contribution.

2）：While the paper examines method performance under various memory and computational constraints, it uses less practical metrics. In real-world applications, memory should be measured in MB or GB, and computation more universally expressed as FLOPs, rather than the paper's approach where "memory means the number of cached samples in edge storage, and computational budget denotes the gradient steps during model update." MB-based memory metrics and FLOPs would provide more meaningful practical insights.

3）：The experimental evaluation primarily focuses on relatively simple datasets such as CIFAR and its variants. Real-world scenarios typically involve more complex and diverse data, including text, speech, and multimodal data. Therefore, the experimental section lacks comprehensive evaluation on more challenging and diverse data types that would better reflect practical applications.

---

> ### Author Rebuttal · Authors · 2025-07-29
>
> > **Q1. Concerns about Novelty**
>
> **R1:** Thank you for your thoughtful comments and suggestions. We fully understand and appreciate the expectation for methodological novelty, and would like to clarify the positioning of our work.
>
> The primary goal of this paper is not to propose a new algorithm, but rather to **conduct the first systematic study of Federated Continual Learning (FCL) under resource-constrained conditions**. While many FCL algorithms have been proposed, there is still a **lack of a unified and standardized evaluation framework** to assess their robustness and generalizability when subject to client-side resource limitations (e.g., memory, computation, and label availability). To address this gap, we establish a large-scale benchmark that models three core resources (Memory/Computation/Label), spans six datasets, includes two task types (Class-IL/Domain-IL), and compares over ten representative methods.
>
> As you rightly pointed out, “such analysis could serve as a foundation for future method development,” and we fully agree. We sincerely appreciate your suggestion to extend this work toward method innovation. In fact, our team has already begun designing **new strategies for resource-constrained FCL** for technical research paper, such as adaptive resource allocation policies that dynamically optimize memory and computation through combinatorial optimization—directly inspired by the empirical insights from this study.
>
> Our current priority is to ensure the analysis is **comprehensive, systematic, and reproducible**, with the hope of benefiting the broader community. In future work, we will build on this benchmark to develop **resource-aware FCL methods**, for which the empirical findings of this study provide a necessary and solid foundation.
>
> > **Q2. (a) Suggest using MB and FLOPs to analyze memory and computation more precisely. (b) The paper should discuss whether similar findings hold under dynamically changing resource constraints.**
>
> **R2:** Thank you for your insightful suggestions. We address your concerns point by point:
>
> (a) We chose to quantify memory (M) by number of samples and computation (B) by number of local gradient steps primarily for the sake of  **fair comparisons** across different baselines in FCL:
>
> - **Memory:** Different datasets vary significantly in sample size. For example, CIFAR-10 samples are 32×32 pixels, while Office-31 includes samples with resolutions up to 4288×2848. Using a fixed unit like megabytes (MB) would result in **inconsistent memory constraints across datasets**, making fair comparison difficult. In contrast, using the **number of samples** directly reflects the **effective memory capacity** for techniques like replay or generation and maintains alignment with their operational semantics.
> - **Computation:** FLOPs depend heavily on model architectures. In our study, some baselines adopt different backbones due to method-specific designs (e.g., FedCIL uses ACGAN, FOT uses AlexNet; see Appendix C). Thus, **FLOPs are not architecture-agnostic** and would introduce bias. We adopt **number of local gradient steps** as a model-independent metric that reflects **actual local computation**: more steps require more local resources regardless of the architecture. Additionally, FLOPs measure **theoretical computation cost** only, while **real-world efficiency** is affected by GPU parallelism, memory bandwidth, data I/O, etc., making FLOPs an **unstable or indirect indicator** of computational burden.
>
> (b) We fully agree that resource constraints in real-world FCL scenarios are often dynamic. For instance, edge devices like smartphones may experience fluctuating compute and memory availability; communication conditions can vary over time, affecting synchronization; labeling frequency and quality may also shift depending on data acquisition or user availability.
>
> Our current study adopts **static resource constraints** with the following goals:
>
> - To **control experimental variables** for a clean analysis of robustness and adaptability under constrained resources;
> - To **establish a standardized resource modeling protocol** as a foundation for future method development;
> - To **quantify the sensitivity and degradation trends** of various algorithms under comparable resource settings.
>
> While we do not simulate dynamic conditions explicitly, some **empirical patterns may indirectly reflect method adaptability**. For example, FedAvg and FOT demonstrate relatively stable performance under tight constraints, suggesting potential suitability for variable environments. Re-Fed’s collaborative replay mechanism can be extended to **dynamically prioritize cached samples** in response to resource fluctuations.
>
> In summary, our key findings—that **the match between resource types and method properties governs performance**—are expected to hold under dynamic conditions, with the dynamics mainly affecting the **amplitude** rather than the **direction** of performance change. We sincerely appreciate your suggestion and will incorporate this important perspective into future work to enhance the practical relevance of our study.
>
> > **Q3. Concerns about Real-world and Multi-Modal Datasets.**
>
> **R3:** We thank the reviewer for the helpful suggestion regarding dataset diversity. We fully agree that a broader range of datasets is crucial for validating the robustness of our conclusions. In this study, we have already included several datasets beyond the CIFAR family to better reflect realistic scenarios:
>
> - **Digit-10** covers handwritten digit images from multiple domains;
> - **Tiny-ImageNet** introduces higher task complexity with 200 diverse categories;
> - **Office-31** and **Office-Caltech-10** contain real-world object images collected across different devices and environments, offering greater heterogeneity in data distribution and domain shift.
>
> These datasets were carefully chosen to ensure diversity in **task complexity** and **data distribution**, thereby enhancing the practical relevance of our benchmark.
>
> Regarding non-vision modalities such as **text** or **speech**, we agree that they represent valuable directions for extending the practical applicability of FCL. However, the primary goal of this work is to establish a **benchmark framework for analyzing FCL under resource constraints**, and most existing FCL methods are still primarily validated on **image-based datasets**. We believe that our current setup sufficiently supports the core conclusions and serves as a solid foundation for future extensions to non-vision domains.

---

> > ### Author Response · Authors · 2025-08-05
> >
> > Dear reviewer ef6U, during this rebuttal period, we have:
> >
> > - Explained the novelty of our empirical study
> > - Clarified the unique contribution and selected limited resources
> > - Explain the datasets and experimental settings
> >
> > In this rebuttal, our responses have received acknowledgment and score improvements from the other two reviewers. We would love to hear your feedback on our updates and look forward to discussing any remaining concerns you may have. Thank you for your time and consideration.

---

> > > ### Comment · Reviewer_ef6U · 2025-08-07
> > >
> > > Thank you for the detailed responses provided in the rebuttal. However, this paper primarily conducts a systematic study of Federated Continual Learning (FCL) under resource-constrained conditions, but does not propose a method that I find sufficiently novel or innovative. Therefore, I believe it does not meet the acceptance bar for NeurIPS, and I will maintain my initial score.

---

> > > > ### Author Response · Authors · 2025-08-08
> > > >
> > > > Thank you for your response. As we stated in the rebuttal, our work is an empirical study on federated continual learning under resource constraints, and it is not intended to propose a new algorithm. We believe that empirical studies are of significant academic value, as they focus on benchmarking, systematic analysis of existing methods, and raising new research questions. Such works are widely recognized by the community and have been published in top-tier conferences. For example, [Prabhu, Ameya, et al. "Computationally budgeted continual learning: What does matter?" Proceedings of the IEEE/CVF Conference on Computer Vision and Pattern Recognition. 2023.] present empirical studies on lightweight continual learning.
> > > >
> > > > Regarding the other concerns you raised, we have already provided detailed responses in the rebuttal. If you have any remaining questions or would like further clarification, please feel free to reach out to us. Your recognition and positive adjustment mean a great deal to us.
> > > >
> > > > Thank you for your valuable suggestions and professional review.
> > > >
> > > > Best wishes,

---

### Official Review · Reviewer_WvFo · 2025-07-04

**Clarity:** 3
**Significance:** 1
**Originality:** 1
**Rating:** 4
**Confidence:** 4

**Summary:**

The paper presents a large-scale empirical evaluation of twelve federated continual learning (FCL) methods under three practical resource constraints: (1) memory buffer budget, (2) computational budget, and (3) label rate (i.e., the fraction of labeled samples available during training). The evaluated methods span four major categories: replay-based methods, regularization-based methods, knowledge distillation, and network extension. The results show that no single FCL method is universally optimal across all resource-constrained settings.

**Questions:**

1. Several non-federated CL works have studied resource constraints extensively. How does this paper build on or differ from them? What challenges are specific to the federated setting that make these constraints harder to handle?
2. Why is communication overhead not considered as part of the resource constraints? Since communication is a major bottleneck in real-world FL systems, an analysis or at least a discussion would improve the paper’s applicability and uniqueness.
3. Could the authors more precisely formalize how each resource constraint is defined and enforced in the experiments? For example, how is computational budget quantified (e.g., gradient steps vs. training time), and how is label rate sampled and applied across clients and tasks?

**Ethical Concerns:**

["NO or VERY MINOR ethics concerns only"]

**Final Justification:**

I would like to thank the authors for their detailed rebuttal, and I apologize for not engaging more during the discussion phase due to an extenuating personal circumstance that required my urgent attention. I have read the rebuttal carefully, along with the other reviewers’ comments and the available discussion. I also had some internal discussion with SBXG, which was useful.

I think the rebuttal and SBXG’s input have addressed my major concerns. However, I still find the contribution limited, since it does not include any specific experiments that show trends that are true in FCL under resource constraints but do not appear in centralized CL under resource constraints. A future paper that does this would be a lot more impactful.

That said, the example of whether insufficient buffer size on clients leads to replay distribution drift that ultimately undermines global knowledge consistency is a good way to distinguish FCL from centralized CL, and I strongly encourage the authors to include it in the paper. I also encourage them to incorporate the clear definitions of resource constraints provided in the rebuttal.

I understand that they initially missed the MICCAI paper, as the field is growing very rapidly, and I hope the references I provided help position the work so that readers can systematically build on relevant CL and FCL results in a systematic way. Finally, I think the future inclusion of ImageNet and DomainNet would make whatever code base published by the authors a lot more attractive.

**Quality:**

2

**Strengths And Weaknesses:**

***Strengths***
* **[Large-scale (in terms of GPU hours) empirical evaluation]**
The paper conducts a large-scale empirical study of federated continual learning methods under three key resource constraints: memory buffer size, computational budget, and label rate. The evaluation spans twelve representative baselines across multiple FCL techniques and datasets, with over 1,000 GPU hours of experiments.
* **[Clarity in organizing techniques and findings]**
The paper organizes FCL methods into four technique categories (sample caching, data synthesis, knowledge distillation, and network extension) and analyzes their performance across varying levels of resource availability. This structured approach makes the findings easier to follow.

***Weaknesses***
* **[Prior work and citation depth]** The paper in its current form lacks discussion of what has been done in the literature. For example, several studies have proposed ways to isolate computational budget and investigate the performance of CL methods when the computational budget is normalized [41, A]. While the paper cites [41], it does not mention how that work evaluates methods under this constraint, nor does it build on any of its insights. Similarly, several works have studied continual learning from partially labeled streams [B, C, D, E], but these are not discussed at all. I understand that these are not federated continual learning papers, but they are still CL papers and highly relevant. Finally, the label rate constraint has already been studied even in a federated continual learning setup, specifically in a medical imaging context [F].
* **[Missing federated-specific challenges]**
Related to the previous point, since all three constraints have been well studied in the CL literature, I expected this paper to focus more on complications that arise specifically from the federated aspect. For example, why is there no discussion of federated communication costs? The paper would benefit from clarifying the specific federated scenario it addresses and clearly delineating how it differs from standard CL, before formalizing the particular resource constraints under study.
* **[Dataset realism and scalability]**
The datasets used are mostly small-scale, and it is unclear whether the findings would generalize to a real-world FCL scenario, which the paper seems to aim to simulate.

[40] Prabhu, Ameya, et al. "Computationally budgeted continual learning: What does matter?." Proceedings of the IEEE/CVF Conference on Computer Vision and Pattern Recognition. 2023.

[A] Ghunaim, Yasir, et al. "Real-time evaluation in online continual learning: A new hope." Proceedings of the IEEE/CVF conference on computer vision and pattern recognition. 2023.

[B] Gomez-Villa, Alex, et al. "Continually learning self-supervised representations with projected functional regularization." Proceedings of the IEEE/CVF Conference on Computer Vision and Pattern Recognition. 2022.

[C] Zhang, Wenxuan, et al. "Continual Learning on a Diet: Learning from Sparsely Labeled Streams Under Constrained Computation." The Twelfth International Conference on Learning Representations.

[D] Csaba, Botos, et al. "Label delay in online continual learning." Advances in Neural Information Processing Systems 37 (2024): 119976-120012.

[E] Caccia, Lucas, and Joelle Pineau. "Special: Self-supervised pretraining for continual learning." International Workshop on Continual Semi-Supervised Learning. Cham: Springer International Publishing, 2021.

[F] Alhamoud, Kumail, et al. "Fedmedicl: Towards holistic evaluation of distribution shifts in federated medical imaging." International Conference on Medical Image Computing and Computer-Assisted Intervention. Cham: Springer Nature Switzerland, 2024.

---

> ### Author Rebuttal · Authors · 2025-07-29
>
> > **W1&Q1.  Unique Nature of Resource Constraints in Federated Continual Learning.**
>
> **R1:** Thank you for the insightful comment. We fully agree that resource constraints have been extensively explored in centralized continual learning (CL). Our work is motivated by these studies, and we would like to clarify the key distinctions and contributions of our work in the context of Federated Continual Learning (FCL):
>
> First, in terms of research focus, prior works often examine the impact of a **single** constraint (e.g., computation budget or labeling rate). In contrast, this paper is, to the best of our knowledge, the first to **systematically investigate the joint effects** of memory buffer size, computation budget, and labeling cost in FCL. We conduct comprehensive evaluations across representative techniques, including generative replay, knowledge distillation, sample-based replay, and regularization-based methods, totaling over 1000 GPU hours.
>
> Second, FCL differs fundamentally from traditional CL in terms of scenario characteristics and research objectives. FCL involves distributed collaborative training among multiple clients with heterogeneous data distributions (simulated via Dirichlet partitions), where privacy constraints prohibit sharing of raw data. Clients must make local decisions based only on limited feedback. Therefore, unlike centralized CL that focuses on optimizing a single model, our goal is to examine **how resource constraints disrupt the collaborative mechanisms unique to FCL**—for example, whether insufficient buffer size on clients leads to replay distribution drift that ultimately undermines global knowledge consistency.
>
> Finally, we thank the reviewer for pointing out [F], which studies label-efficiency in a FCL.  **MICCAI is a top-tier conference in the medical domain, and related methods primarily emphasize applications to medical tasks, whereas our work focuses on more general algorithmic research from the ML/CV/AI community.** Due to this difference in focus, we were previously unaware of [F]. This is highly relevant, and we will cite and discuss it properly.
>
> > **W2&Q2.  Exploration of Communication Cost.**
>
> **R2:** Thank you for your insightful comments. We fully agree that communication overhead is one of the core challenges in FL, especially in real-world deployments. In this work, we did not explicitly include communication cost as one of the three core resource constraints, mainly due to the following considerations:
>
> **Communication cost is fundamentally different from the client-side training constraints we focus on.** Our study primarily targets three types of **client-local resource limitations**: memory buffer, computational budget, and label rate. These are hard constraints that directly affect the training capacity of edge devices. In contrast, communication cost is mainly determined by factors such as communication frequency, model synchronization strategy (e.g., number of rounds $T$), and model size. **Therefore, it impacts training latency rather than the local training capability of clients.**
>
> **Communication constraints deserve dedicated treatment in future studies.** We appreciate your point that communication cost is an important dimension in FCL. For example, future studies could explore communication heterogeneity (e.g., bandwidth and latency across clients), the applicability of compression/sparsification techniques in FCL, or trade-offs between communication and local computation. As noted in Appendix E, we have already listed communication constraints as a direction for future work. In the current paper, we aim to first provide a systematic understanding of client-side resource limitations, which we believe is a necessary step toward building a more comprehensive performance analysis framework for FCL.
>
> > **W3. Concerns about Datasets.**
>
> **R3:** Thank you for raising the concern regarding dataset scale. We fully understand and agree that large-scale datasets are important for simulating realistic FCL scenarios. We address your comments as follows:
>
> **1. The selected datasets are common datasets in FCL research, ensuring representativeness and fairness**
>
> In this work, we evaluate our method on six publicly available datasets (CIFAR-10/100, Tiny-ImageNet, Digit-10, Office-31, and Office-Caltech-10), which are widely adopted in existing FCL literature to assess algorithmic performance under class-incremental (Class-IL) and domain-incremental (Domain-IL) settings. These datasets offer the following advantages:
>
> - They cover **diverse application scenarios**, including natural images (CIFAR, Tiny-ImageNet), handwritten digits (Digit-10), and cross-domain settings (Office series);
> - They span **a range of class complexities**, from 10 to 200 classes, enabling meaningful evaluation of interference and forgetting during continual learning;
> - They are **easy to reproduce and widely used**, facilitating fair and standardized benchmarking.
>
> **2. We plan to extend the evaluation to larger FCL datasets such as ImageNet and DomainNet**
>
> We greatly appreciate your suggestion to further validate the scalability of our method. We plan to expand our experiments to large-scale datasets like **ImageNet** and **DomainNet** in future work. Due to the significant training cost, we are currently unable to provide full-scale results during the rebuttal phase, but we will prioritize this direction in our subsequent research.
>
>
> > **Q3. Clear Definition of Resource Constraint.**
>
> **A3:** Thank you for your question regarding the precise definition and application of resource constraints. Below, we formalize the three key constraints considered in our work — memory buffer (M), computational budget (B), and label rate (R) — and explain how they are consistently enforced in our federated continual learning (FCL) experiments.
>
> **1. Memory Buffer (M)**
>
> - **Definition**:
>   $M \in \mathbb{N}$ represents the maximum number of samples that a client can store from **previous tasks**. This buffer may include:
>
>   - real samples from earlier tasks,
>   - synthetic samples generated by a local generative model,
>   - or samples used for distillation-based replay.
>
> - **Enforcement**:
>   At task $t$ , each client builds its local training set as:
>   $$
>   D_\text{train}^{(t)} = D_\text{current}^{(t)} \cup D_\text{buffer}^{(t-1)}, \quad \text{with} \quad |D_\text{buffer}^{(t-1)}| \leq M
>   $$
>   When the cache data exceeds $M$, we retain $M$ samples through random sampling.
>
>   This ensures that the memory usage for prior knowledge remains strictly bounded.
>
> **2. Computational Budget (B)**
>
> - **Definition**:
>   $B \in \mathbb{N}$ denotes the maximum number of **gradient update steps (i.e., backward passes)** a client may perform **during local training in total  communication rounds**. This constraint is **applied only on the client side**, since server-side resources are typically not the bottleneck in real-world FCL systems.
>
>   This budget includes **all client-side components**, such as:
>
>   - the main local model,
>   - local generative models,
>   - local personalized models,
>   - and distillation losses.
>
> - **Enforcement**:
>   In PyTorch, we track the number of times the `.backward()` function is called during local training. Once a client reaches or exceeds $B$ calls to `backward()`, its local training is immediately terminated:
>
>   ```python
>   if backward_calls >= B:
>       break  # Stop training for this client
>   ```
>
>   This unified constraint ensures fair and consistent computational load across different methods.
>
> **3. Label Rate (R)**
>
> - **Definition**:
>   $R \in (0, 1]$ defines the **fraction of labeled data** available to each client per task.
>
> - **Enforcement**:
>   For a given task $t$, each client samples a labeled subset from its local data:
>   $$
>   D_t = D_t^{\text{labeled}} \cup D_t^{\text{unlabeled}}, \quad \text{with} \quad |D_t^{\text{labeled}}| = R \cdot |D_t|
>   $$
>   Only $D_t^{\text{labeled}}$  is used for loss computation.

---

> > ### Author Response · Authors · 2025-08-05
> >
> > Dear reviewer WvFo, during this rebuttal period, we have:
> >
> > - Analyze the unique nature of resource constraints in federated continual learning
> > - Explain the reason for the selected datasets.
> > - Explain the reason why we excluded the communication cost
> > - Clarify definitions of each resource constraint.
> >
> > In this rebuttal, our responses have received acknowledgment and score improvements from the other two reviewers. We would love to hear your feedback on our updates and look forward to discussing any remaining concerns you may have. Thank you for your time and consideration.

---

### Note · Authors · 2025-08-13

Dear SACs, ACs, Reviewers,

Thanks for all your effort in reviewing our paper submitted to NeurIPS! Considering the large variance in scores, which may confuse the AC’s judgment, we would like to provide a brief final clarification.

First, we have concerns that reviewer WvFo may not have fully fulfilled their reviewing responsibility. The reviewer’s comments emphasized progress in continual learning, whereas our work focuses on an empirical study of federated continual learning under resource constraints, in which we also propose and analyze unique challenges specific to this distributed setting.

Moreover, for an empirical study, it is both necessary and valuable to unify different methods under a consistent framework and to conduct extensive experiments on relevant baselines. Such studies play an important role in the research community by uncovering the limitations of existing approaches and fostering further discussion. As we pointed out in the rebuttal, several works of this type have been published at top-tier conferences, and it is unnecessary to propose a new method.

Finally, we respectfully request that the AC take into account the issues we previously raised during the rebuttal phase when making the final decision, in order to ensure a fair and accurate review process.

Thank you for your time and effort in managing the review process.

Best regards,

Authors

---

### Decision · Program_Chairs · 2025-09-17

**Decision:**

Accept (poster)

**Comment:**

This paper provides a large-scale empirical study of federated continual learning (FCL) under resource constraints, including memory, computation, and label availability. By benchmarking a broad set of methods across six datasets and two task types, the authors show that existing approaches degrade significantly in constrained settings and offer insights into their limitations.

The work's strengths are its scope and relevance: over 1,000 GPU hours of experiments, a structured comparison across method families, and clear takeaways that distinguish FCL-specific challenges from centralized continual learning. One reviewer highlighted that systematic empirical studies are valuable for advancing the field.

The main weaknesses suggested were the lack of algorithmic novelty, limited comparison to CL under resource constraints (i.e., what makes FCL different?), reliance on relatively small-scale datasets, and omission of communication cost as a core resource. Some reviewers also found the chosen metrics (samples and gradient steps) less practical than MB or FLOPs.

Nonetheless, concerns were partly addressed in rebuttal and discussion. Reviewers who initially leaned negative acknowledged the clarifications and raised scores, while another was satisfied with revisions on clarity. The final balance of reviews supports acceptance, given the careful experimental depth and design.

For camera-ready, the authors should make sure to include updates per the discussion and final reviews (e.g., including the example of insufficient buffer size). The idea of incorporating communication constraints into the evaluation should be added in the discussion section. All other discussions should be incorporated or clarified in the final version.